# Female genital schistosomiasis burden and risk factors in two endemic areas in Malawi nested in the Morbidity Operational Research for Bilharziasis Implementation Decisions (MORBID) cross-sectional study

Olimpia Lamberti[1]*, Sekeleghe Kayuni[2,3,4,5], Dingase Kumwenda[2], Bagrey Ngwira[1†], Varsha Singh[6], Veena Moktali[6], Neerav Dhanani[7], Els Wessels[8], Lisette Van Lieshout[9], Fiona M. Fleming[7], Themba Mzilahowa[2], Amaya L. Bustinduy[1]*

1 Department of Clinical Research, London School of Hygiene and Tropical Medicine, London, United Kingdom, 2 Centre for Health, Agriculture and Development Research and Consulting (CHAD), Blantyre, Malawi, 3 MASM Medi Clinics Limited, Medical Aid Society of Malawi (MASM), Lilongwe, Malawi, 4 Malawi Liverpool Wellcome Programme (MLW), Kamuzu University of Health Sciences (KUHeS), Queen Elizabeth Central Hospital (QECH), Blantyre, Malawi, 5 Department of Tropical Disease Biology, Liverpool School of Tropical Medicine, Liverpool, United Kingdom, 6 Periwinkle Technologies Pvt Ltd, Pune, India, 7 Unlimit Health, London, United Kingdom, 8 Department of Medical Microbiology, Leiden University Medical Center, Leiden, The Netherlands, 9 Department of Medical Parasitology, Leiden University Medical Center, Leiden, The Netherlands

† Deceased.

* Olimpia.Lamberti@lshtm.ac.uk (OL); Amaya.bustinduy@lshtm.ac.uk (ALB)

## Abstract

### Background

Female genital schistosomiasis (FGS), caused by the parasite *Schistosoma haematobium (Sh)*, is prevalent in Sub-Saharan Africa. FGS is associated with sexual dysfunction and reproductive morbidity, and increased prevalence of HIV and cervical precancerous lesions. Lack of approved guidelines for FGS screening and diagnosis hinder accurate disease burden estimation. This study evaluated FGS burden in two *Sh*-endemic areas in Southern Malawi by visual and molecular diagnostic methods.

### Methodology/Principal findings

Women aged 15–65, sexually active, not menstruating, or pregnant, were enrolled from the MORBID study. A midwife completed a questionnaire, obtained cervicovaginal swab and lavage, and assessed FGS-associated genital lesions using hand-held colposcopy. '*Visual-FGS*' was defined as specific genital lesions. '*Molecular-FGS*' was defined as *Sh* DNA detected by real-time PCR from swabs. Microscopy detected urinary *Sh* egg-patent infection.

In total, 950 women completed the questionnaire (median age 27, [IQR] 20–38). Visual- and molecular-FGS prevalence were 26·9% (260/967) and 8·2% (78/942), respectively. 6·5% of women with available genital and urinary samples (38/584) had egg-patent *Sh*

**Data Availability Statement:** All relevant data are available at https://doi.org/10.17037/DATA.00003850.

**Funding:** This work received financial support from the Coalition for Operational Research on Neglected Tropical Diseases, which is funded at The Task Force for Global Health through the United States Agency for International Development. Grant number: NTD-SC 158.3D.The funders had no role in study design, data collection and analysis, decision to publish, or preparation of the manuscript.

**Competing interests:** The authors have declared that no competing interests exist.

infection. There was a positive significant association between molecular- and visual-FGS (AOR = 2·9, 95%CI 1·7–5·0). 'Molecular-FGS' was associated with egg-patent *Sh* infection (AOR = 7·5, 95% CI 3·27–17·2). Some villages had high 'molecular-FGS' prevalence, despite <10% prevalence of urinary *Sh* among school-age children.

## Conclusions/Significance

Southern Malawi carries an under-recognized FGS burden. FGS was detectable in villages not eligible for schistosomiasis control strategies, potentially leaving girls and women untreated under current WHO guidelines. Validated field-deployable methods could be considered for new control strategies.

## Author summary

Female genital schistosomiasis (FGS) is a neglected gynaecological disease caused by the waterborne parasite *Schistosoma (S.) haematobium*. Despite over 45 million women are at risk of FGS in sub-Sahara Africa (SSA), approximately only 15,000 have been screened for the disease. Diagnosis is challenging and has traditionally required high technical expertise based on visual inspection for FGS typical lesions of the genital tract using a standard colposcope, seldom available in endemic settings. Closer-to the-user and decentralized strategies for FGS screening and diagnosis should be implemented to assess disease burden and scale-up FGS surveillance. This study was nested within the larger Morbidity Operational Research for Bilharziasis Implementation Decision (MORBID) cross-sectional project, aiming to correlate schistosomiasis-related morbidity data with village level endemicity across two districts in Southern Malawi. We found a significantly moderate to high burden of FGS (between 8–27% depending on diagnostic method used), with marked age differences in diagnostic performance. Further, some villages with low schistosomiasis prevalence (which would be excluded from control strategies per new WHO guidelines), had a significantly high burden of FGS, indicating the need for formal public health interventions. Within the remit of the sustainable development goals, this study's approach and findings emphasize the need of a field-deployable strategy to FGS screening and diagnosis in endemic areas in Malawi and other similar setting.

## Introduction

Female genital schistosomiasis (FGS) is a neglected gynaecological condition caused by the parasite *Schistosoma (S.) haematobium* [1,2]. FGS is highly prevalent in Sub-Saharan Africa (SSA) and is associated with poor reproductive outcomes, including infertility, abortion, and ectopic pregnancies, with negative social and psychological impacts [3]. There is also emerging evidence of an increase in HIV and cervical precancerous lesions in women with FGS [1,4]. Early diagnosis and prevention of FGS are essential to achieve the aspirations of the Sustainable Development Goals (SDGs) on health promotion and women's empowerment [5].

Awareness of FGS is largely absent in *S. haematobium*-endemic countries despite the negative outcomes associated with the disease [2,6]. Clinical manifestations of FGS include vaginal discharge, vaginal itching, and abdominal pain, symptoms which are often mistakenly attributed to sexually transmitted infections (STIs) by healthcare workers and sufferers alike [2].

FGS diagnosis is challenging as there is not a reference standard for screening and diagnosis [1,2,7]. Histological examination of cervical tissues obtained by biopsy from a suspicious genital lesion can identify *S. haematobium* eggs in the genital tract and is considered the gold standard for FGS diagnosis [8,9]. Histopathological studies for FGS diagnosis, however, are scarce mostly due to lack of pathology services and hypothetical concerns of increased risk of HIV transmission in high endemicity areas [1]. Standard diagnosis is based on visual inspection for FGS-typical lesions on the cervix or vaginal walls using a colposcope [1,2]. Images are classified as suggestive of '*visual-FGS*' if homogeneous yellow sandy patches, grainy sandy patches, abnormal blood vessels, or rubbery papules are observed [10]. However, colposcopy requires good infrastructure, costly equipment, and high-level specialized training, all seldom available in rural settings where *S. haematobium* is endemic [1,2]. Importantly, FGS diagnosis from cervicovaginal images may lack specificity, limiting its diagnostic accuracy [1,11]. The FGS mucosal changes visually observed with colposcopy are non-specific and have also been associated with other genital infections and STIs [11]. This can lead to over-treatment of FGS and overlooking other infections [11]. In addition, visual diagnosis relies on the imperfect human expert review of images, which is highly subjective with low inter-rater correlation across reviewers [11]. Overall, these limitations hinder the ability to obtain accurate FGS prevalence estimates [1]. Molecular diagnostic methods such as the polymerase chain reaction (PCR) for parasite DNA detection from genital samples, collected either at home or in clinic, have been validated in a study in Zambia and are being used in different countries, as an alternative, more accurate and less invasive method of FGS diagnosis in field settings [2,12,13]. To the best of our knowledge, no study has assessed molecular methods for FGS screening and diagnosis in Malawi.

Previously, a systematic review of studies found that *S. haematobium* infections in Malawi are highly localized in the Southern Lake region [14–16]. A study at Zomba hospital (Malawi) between 1974–75 detected *Schistosoma* ova from cervical biopsies in 43·5% (60/138) of women who presented infertility symptoms [17]. A later study (1976–80) reviewed the histopathology of 176 cases of known gynaecological schistosomiasis in Blantyre, Malawi and found that 60·0% of cases involved the cervix [18]. Further, a study in 1996 showed that 65·0% (33/51) of women with *S. haematobium* in the urine presented ova in the cervix, vagina, and vulva by genital biopsies [19].

To the best of our knowledge, no FGS study has been conducted in Malawi in the last 25 years. This limits the estimation of the FGS actual prevalence and hinders the development of effective control strategies. This study aimed to fill this gap by estimating the FGS burden by conventional and novel diagnostic methods in two *S. haematobium* endemic areas in Malawi as part of the larger 'Morbidity Operational Research for Bilharzia Implementation Decisions' (MORBID) study, which aims to compare schistosomiasis-associated morbidity between low- and high-level infection prevalence communities. This study also assessed the urinary, and sexual and reproductive health (SRH) symptoms associated with FGS.

## Methods

### Ethical considerations

The study was approved by the National Health Sciences Research Committee (NHRSC) in Malawi (Approval number: 2404) and the London School of Hygiene and Tropical Medicine (Approval number 19072 /RR/17763)

### Study design

Between November 2020 and May 2021, 1,000 girls and women aged 15–65 years old, registered in the MORBID study, were randomly selected, using simple random sampling, to

participate in the present cross-sectional MORBID For Girls and Women (MORBID-FGS) pilot study (S4 Text). To be eligible to participate, individuals had to be aged 15–65 years, non-pregnant, and sexually active.

The MORBID study was a cross-sectional community-based study to assess the level of *S. haematobium* specific morbidity markers across Nsanje and Chikwawa districts in Southern Malawi (S1 Text). Sixty villages per district were randomly selected, half from high and low prevalence areas, determined by epidemiological mapping reassessment surveys (2017–2019) (S2 Text). In each village, 50 individuals from each age groups, including pre-school aged children (2–6 years old), school aged children (7–13 years old), adolescents (14–19 years old), and adults (aged over 20 years old), were randomly selected using household-level random sampling (200 individuals in total) (S3 Text). As part of the MORBID study, participants provided a single urine sample to estimate egg-patent *S. haematobium* infection prevalence and intensity through urine filtration (S5 Text).

## Exposure covariates

After obtaining written informed consent, participants of the MORBID-FGS study completed a field-validated questionnaire in either English or the national language (Chichewa), administered by a midwife at the local clinic. Questions included socio-demographic characteristics, water contact information, and history of urinary, genital, and SRH signs and symptoms.

## FGS diagnosed by visual inspection

During the same clinic visit, a midwife captured images of the cervix, fornices and vagina using a hand-held colposcope (EVA MobileODT). Additional images were collected on a subgroup of participants using the "Smart-Scope" hand-held device [20]. Images were evaluated remotely by the study gynaecologist (DK) in Malawi and classified as '*visual-FGS*' if homogeneous sandy patches, grainy sandy patches, rubbery papules, or abnormal blood vessels were present, and negative if none were observed [10]. Participants with any suspicious lesions were referred to the study gynaecologist for follow-up and treatment as per national Malawi guidelines [21].

## FGS diagnosed by molecular methods

In the clinic, midwives also collected a cervicovaginal swab using a Dacron swab commonly used in the Malawi health system (S1 Fig). After speculum insertion, the swab was inserted vaginally and rotated at 360 degrees. Swabs were placed in individual screw cap microtubes, stored in PrimeStore (STARLAB, Hamburg, Germany) and placed in a refrigerator in the laboratory before shipping to Leiden University Medical Center (LUMC), in the Netherlands for DNA extraction and real-time PCR analysis. Afterwards, a cervicovaginal lavage (CVL) was obtained. After speculum insertion, normal saline (10mL) was flushed with a syringe over the anterior cervix and vaginal walls for one minute. The lavage was then collected from the posterior fornices with a pipette and placed in a centrifuge tube with PrimeStore [2].

All specimens were tested by PCR at LUMC. Samples were vortexed and the PrimeStore solution was transferred to a 2 mL tube containing Precellys Soil grinding SK38 (Bertin technology, Montigny-le-Bretonneux, France). DNA extraction used the MagNA Pure 96 system (Roche Diagnostics, Penzberg, Germany) (S6 Text). Schistosoma-specific real-time PCR was performed as previously described [2]. DNA amplification and detection were performed with the CFX-96 Real Time PCR Detection System (BioRad, California, USA). Output quantification cycles (Ct-value) was analysed using BioRad CFX software. Any Ct-value observed was classified as '*molecular-FGS*' positive (S6 Text). Workers at LUMC performing PCRs were blinded to clinical and microscopy data [2,22]. Women were treated with a single dose of

praziquantel at 40mg/kg as per WHO guidelines if evidence urinary *S. haematobium*, '*visual-FGS*' or '*molecular-FGS*' was found [21].

## Statistical analysis

Visual and molecular FGS status was matched with the MORBID-FGS questionnaire data using unique individual identifiers (IDs). A subset dataset including urinary *S. haematobium* infection status was made by matching IDs and age between MORBID-FGS and the main MORBID datasets. Age was used as an additional variable to ensure accurate matching when handling duplicates.

Data were analysed using STATA 17·0. The outcomes were '*visual-FGS*' and '*molecular-FGS*' which correlated to visual diagnosis of image taken by hand-held colposcopy and to *S. haematobium* DNA detection by PCR of genital samples, respectively. Multivariable logistic regression models were used to understand the association between risk factors and the two outcomes of interest. Covariates were then considered as significant risk factors if they were significantly associated with the outcomes in univariable analyses tested using Pearson's chi-square ($\chi^2$) and Wilcoxon-Mann-Whitney tests (S1 Table). Models were then built by adding one variable at the time and tested using likelihood ratio tests.

Associations between molecular or visual FGS and individual-level urinary *S. haematobium* egg-patent infection was assessed with generalized linear mixed models with binomial link function on the subset. Explanatory variables included *S. haematobium* status, measured in the main *parent* MORBID study, and standardized age. Village was treated as a random effect.

Simple liner regression models were used to explore the association of FGS with urinary *S. haematobium* village-level prevalence in school-aged children (SAC), expressed as a continuous numerical outcome. Explanatory variables included '*molecular-FGS*' by PCR of genital samples, and '*visual-FGS*' by EVA MobileODT and Smart-Scope. Only villages with at least 15 observations were included.

## Results

Fig 1 shows the study flow. Overall, 1,015 women from the MORBID population cohort met the inclusion criteria and 994/1,015 were included in MORBID-FGS study (Fig 1). 950/994 completed the questionnaire. Hand-held colposcopy was performed on 967/994 women using EVA MobileODT and on 448/967 using Smart-Scope. Cervicovaginal swabs were collected

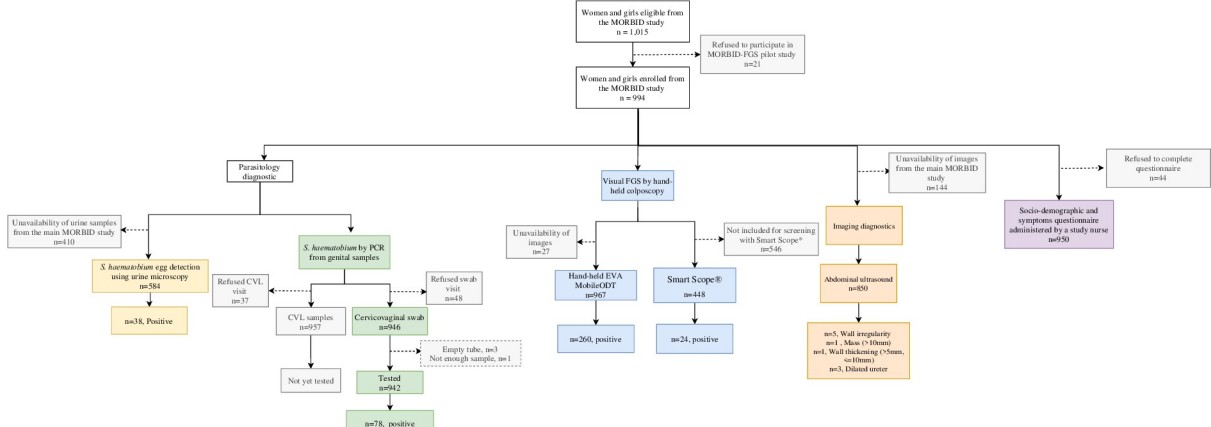

**Fig 1. Flowchart for participant recruitment of 994/1,015 girls and women from the MORBID study who underwent urine microscopy, urinary tract ultrasound, genital PCR, and colposcopy.**

from 946/994 women and 942/946 were analysed by PCR. 'Visual-FGS' and 'molecular-FGS' status were available on 842/950 women who completed the questionnaire. Among participants with visual and molecular FGS observations, 537 and 544 had urinary samples available, respectively. Some participants did not complete the questionnaire or screening procedures, resulting in different denominators across procedures.

### Baseline characteristics of the study population

The median age across the study population was 27 years old (IQR 20–38 years-old). 85·5% (679/950) of women reported taking more than one year to get pregnant. 9·0% (85/950) of women reported previous diagnosis of STI (S2 Table)

### Prevalence of *S. haematobium* infection

Urine samples were available from 584 women and the prevalence of egg-patent urinary *S. haematobium* infection was 6·5% (38/584). Prevalence was higher in Nsanje than Chikwawa ($\chi^2$ p-value = 0·02) (Table 1).

### Self-reported signs and symptoms by FGS status

Participants self-reported abdominal and genital pain, haematuria, difficulty passing urine, vaginal itching, vaginal bleeding after sexual intercourse, amenorrhea, and fear of pain during intercourse. Amenorrhea was the only sign associated with 'visual-FGS' with higher prevalence in women without 'visual-FGS', compared to women with 'visual-FGS' (S2 Fig and S5 Table). Self-reported urinary and SRH symptoms were not significantly associated with the typical FGS cervicovaginal lesions in 'visual-FGS' positive women (S6 Table). Haematuria was the

**Table 1. Diagnostic test results and prevalence of 'visual-FGS, 'molecular-FGS', and urinary *S. haematobium* infection across the study population and by the two study districts in Southern-Malawi.**

| "*Visual-FGS*" positive by different lesions* (Mobile ODT EVA hand-held colposcopy) | | Overall prevalence** (n = 880) N(%) | Chikwawa (n = 374) N(%) | Nsanje (n = 506) N(%) | p-value+ |
|---|---|---|---|---|---|
| | Overall prevalence of "*Visual-FGS*" | 247(28·1%) | 49 (13·1%) | 198 (39·1%) | <0·001 |
| | Homogeneous yellow sandy patches | 207 (23·5%) | 39 (10·4%) | 168 (33·2%) | <0·001 |
| | Grainy sandy patches | 31(3·5%) | 3(0·8%) | 28(5·5%) | <0·001 |
| | Rubbery papules | 7 (0·8%) | 0 | 7(1·3%) | <0·001 |
| | Abnormal blood vessels | 15 (1·7%) | 1 (0·3%) | 14 (2·8%) | <0·001 |
| "*Molecular-FGS*" positive (PCR on cervicovaginal swabs) | | **Overall prevalence** (n = 909) N(%)** | **Chikwawa (n = 375) N(%)** | **Nsanje (n = 534) N(%)** | **p-value+** |
| | Prevalence | 68(7·5%) | 20(5·3%) | 48(9·0%) | 0·04 |
| | Median Ct (IQR) | 35·3 (27·8–38·1) | 36·7 (33·7–38·8) | 33·5 (26·5–37·4) | |
| *S· haematobium* egg-patent infection (urine microscopy) | | **Overall prevalence** (n = 584) N(%)** | **Chikwawa (n = 259) N(%)** | **Nsanje (n = 325) N(%)** | **p-value+** |
| | Prevalence | 38(6·5%) | 10(3·9%) | 28(8·6%) | 0·02 |
| | Mean number (SD) of eggs/10mL urine | 3·4 (27·2) | 1·7 (18·9) | 4·6 (32·0) | |

*Overall numbers for Smart scope are available in S9 Table

** 'Overall prevalence' refers to the prevalence across the study population and was calculated after matching FGS status with observations from the MORBID-FGS questionnaire data

+P-values were calculated using Pearson chi-squared test for the comparison of Chikwawa and Nsanje

Percentages are calculated as the proportion of positive cases by the total number of observations (i.e. the denominator is the number N from the columns)

Participants can present with multiple cervicovaginal lesions at the same time. The sum of observations across columns is therefore larger than the total number of observations for 'visual-FGS'.

only sign associated with *'molecular-FGS'* showing higher prevalence in women without *'molecular-FGS'*, compared to women with it (S5 Table).

## Visual-defined FGS

The prevalence of *'visual-FGS'* by EVA MobileODT across the study population was 26·9% (260/967). After matching *'visual-FGS'* data with the MORBID-FGS questionnaire data, *'visual-FGS'* status was available for 91·0% (880/967) of women. The prevalence of *'visual FGS'* in this subset was 28·1% (247/880) (Table 1). Urinary *S. haematobium* status was available for 145/260 (55·7%) of women with *'visual-FGS'* and the prevalence of egg-patent urinary schistosome infection was 8·3% (12/145) ($\chi^2$ p-value = 0·56). Of the 38/584 women with egg-patent infection, 31·5% (12/38) had *'visual-FGS'* by EVA MobileODT hand-held colposcopy.

The prevalence of *'visual-FGS'* by Smart-Scope assessment was 5·4% (24/448). *'Visual-FGS'* status by EVA MobileODT colposcopy was available for 427/448 (95·5%) of women assessed with Smart-Scope. The Cohen's kappa statistics for diagnosis agreement between the hand-held colposcopes was 0·09, indicating "slight" agreement (p-value = 0·02). The two colposcopes had 1·4% (6/427) and 83·6% (358/427) agreement on positive and negative diagnosis, respectively. 14·7% (63/427) of cases had a discordant diagnosis of *'visual-FGS'*.

## Molecular-defined FGS

The prevalence of *'molecular-FGS'* across the study population was 8·2% (78/942). After matching the *'molecular-FGS'* and MORBID-FGS questionnaire datasets, 96·5% (909/942) observations for *'molecular-FGS'* were available. In this subset, the prevalence of *'molecular-FGS'* was 7·5% (68/909) (Table 1). *'Molecular-FGS'* data was available from 241/260 women with *'visual-FGS'* (93·8%). Among the women with *'visual-FGS'*, 15·4% (37/241) were *'molecular-FGS'* positive ($\chi^2$ p-value<0·001 molecular- vs. visual-FGS). Of the 78 women with *'molecular-FGS'*, 72 (92·3%) were also diagnosed for *'visual-FGS'* and 51·4% (37/72) were *'visual-FGS'* positive ($\chi^2$ p-value<0·001 molecular- vs. visual- FGS). S6 Table shows the prevalence of *'molecular-FGS'* by lesion type. 48·7% (38/78) of women with *'molecular-FGS'* were also tested for urinary *S. haematobium* infection and the prevalence of active urinary schistosome infection was 28·9% (11/38) ($\chi^2$ p-value<0·001 *'molecular-FGS'* vs. urinary *S. haematobium*). Of the 60 women with active schistosome infection, 28·9% (11/38) had *'molecular-FGS'*.

## Distribution of infection by age

Women aged 50+ had higher mean prevalence of *'visual-FGS'* compared to younger women ($\chi^2$ p-value = 0·001 for *'visual-FGS'* across age-groups) (Fig 2). No statistically significant difference of *'molecular*-FGS' was found across age groups ($\chi^2$ p-value = 0·05 for *'molecular-FGS'* across age) (Fig 2).

## Risk factors associated with *'visual-FGS'* and *'molecular-FGS'*

Tables 2 and 3 report the adjusted odds ratios (AOR) for *'visual-FGS'* status and *'molecular-FGS'* status in 842 women with complete data. Women aged 50+ years-old were more likely to have *'visual-FGS'* compared to women aged 15–20 years-old (AOR = 3·1, 95%CI 1·48–5·43, p-value = 0·002). Younger women, aged 15–30 years-old, had higher odds of having *'molecular-FGS'* compared to women aged 41–50 years-old (15–19 years-old: AOR = 5·5, 95%CI 1·20–24·7, p-value = 0·03; 20–30 years old: AOR = 5·7, 95% CI 1·32–24·2, p-value = 0·02). Women with *'molecular-FGS'* were more likely to also be positive for *'visual-FGS'* compared to *'molecular-FGS'* negative women (AOR = 2·9, 95% 1·67–4·95, p<0·001).

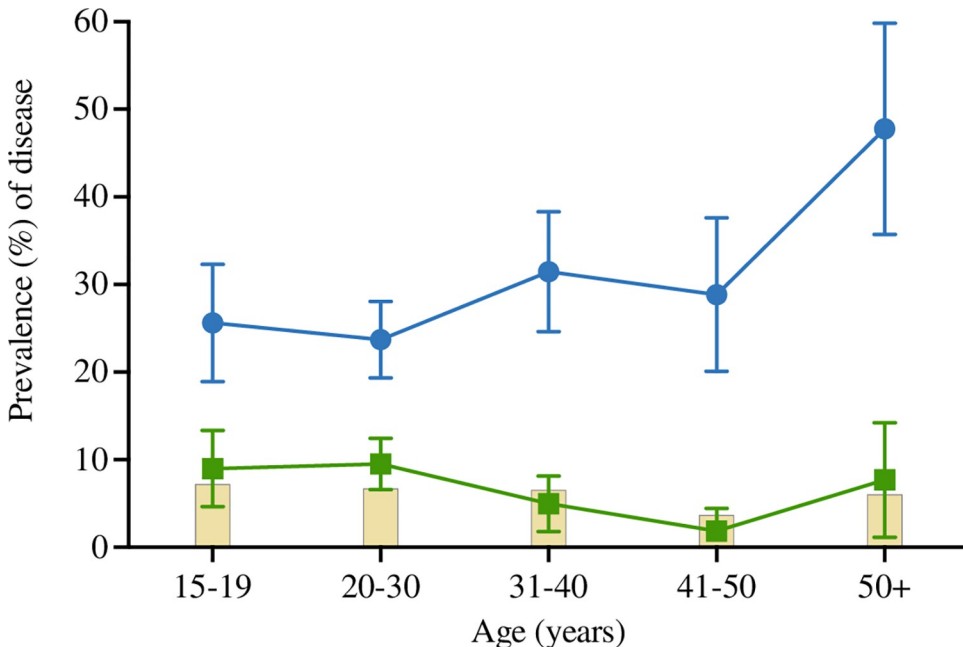

**Fig 2. Distribution of *'visual FGS'* by EVA MobileODT hand-held colposcopy and *'molecular FGS'* prevalence across age groups.** The corresponding data are shown in S8 Table. *'Visual-FGS'* status was significantly associated with age in univariable analysis (Pearson's chi-squared p-value = 0·001). * *'Visual-FGS'* status was significantly different (p-value<0·05) in women in the 50+ years-old age group compared to other age-groups. No statistically significant difference between *'molecular-FGS'* and age groups was observed.

## Association between FGS status and urinary *S. haematobium* infection

Women with positive urinary *S. haematobium* infection had 7·5 higher odds of positive diagnosis for *molecular-FGS* compared to those without (AOR = 7·5, p-value<0·001) (S10 Table).

**Table 2. Multivariable logistic regression model parameter estimates with *'visual FGS'* by EVA MobileODT as the dependent variable in the study population (n = 842).**

| Variables | | Multivariable model | |
|---|---|---|---|
| | | AOR (95% CI)[1] | p-value |
| Age (years) | 15–19 | 1 | |
| | 20–30 | 0·9 (0·6–1·4) | 0·62 |
| | 31–40 | 1·5 (0·9–2·5) | 0·15 |
| | 41–50 | 1·7 (0·9–2·9) | 0·13 |
| | 50+ | 3·1 (1·5–5·4) | 0·002 |
| District | Chikwawa | | |
| | Nsanje | 3·9 (2·7–5·7) | <0·001 |
| History of STI | No | 1 | |
| | Yes | 0·4 (0·2–0·8) | 0·02 |
| *'Molecular-FGS'* by PCR | No | 1 | |
| | Yes | 2·9 (1·7–5·0) | <0·001 |

[1]AOR = Adjusted Odds Ratio, CI = Confidence Interval

Model adjusted for age, district, history of sexually transmitted infections (STI) and *'Molecular FGS'* status.

Confounding variables were selected based on a review of the literature and statistical significance from the univariable association with the outcome

**Table 3. Logistic regression model parameter estimates with *'molecular-FGS'* diagnosed by genital PCR as the dependent variable in the study population (n = 842).**

| Variables | | Multivariable model | |
|---|---|---|---|
| | | AOR (95%CI)[1] | p-value |
| Age (years) | 15–19 | 5·5 (1·2–24·7) | 0·03 |
| | 20–30 | 5·7 (1·3–24·2) | 0·02 |
| | 31–40 | 2·6 (0·5–12·2) | 0·23 |
| | 41–50 | 1 | |
| | 50+ | 3·3 (0·6–17·5) | 0·17 |
| District | Chikwawa | 1 | |
| | Nsanje | 1·3 (0·8–2·4) | 0·32 |
| *'Visual-FGS'* by colposcopy | Negative | 1 | |
| | Positive | 2·9 (1·7–5·0) | <0·001 |

[1]AOR = Adjusted Odds Ratio, CI = Confidence Interval

Model adjusted for age, district and '*Visual FGS*' status.

Confounding variables were selected based on a review of the literature and statistical significance from the univariable association with the outcome

There was no significant association between *visual-FGS* and *S. haematobium* infection status (p-value = 0·5) (S11 Table).

30 villages had sufficient observations of '*molecular-FGS*' and '*visual-FGS*' (EVA MobileODT), and 15 had sufficient observations of *visual-FGS* by Smart-Scope. There was a weak but significant positive correlation between village-level *S. haematobium* prevalence in SAC and '*molecular-FGS*' prevalence (r-squared = 0·31, p = 0·04) (Fig 3A). '*Molecular-FGS*' prevalence was high in some low-prevalence villages for *S. haematobium* in SAC (<10%) [21] (Fig 3A). No significant correlation between '*visual-FGS*' and urinary egg-patent *S. haematobium* prevalence among village SAC was found (r-squared = 0·04, p = 0·23 for EVA MobileODT; r-squared = -0·02, p = 0·40 for Smart-Scope).

## Discussion

To the best of our knowledge, the present MORBID-FGS study is the first to assess the burden and risk factors of FGS in Malawi using visual and molecular diagnostic methods. To date, this is the largest cross-sectional study conducted on FGS, including over 950 women. The baseline prevalence of '*visual-FGS*' across the study population was higher than '*molecular-FGS*', with a direct age association. Importantly, we showed a strong significant association between visual and molecular diagnostic methods for FGS, after adjusting for confounders. This large study found no statistically significant associations between SRH self-reported symptoms and FGS status (both visual and molecular). Prevalence of '*molecular-FGS*' was high in villages with SAC infection prevalence below 10%, as detected by urine filtration.

Across our study population, '*visual-FGS*' prevalence was higher than prevalence of '*molecular-FGS*' (26·9% versus 8·2%). The mean prevalence of '*visual-FGS*' significantly increased with age and peaked in women aged 50+. In contrast, more mid-younger women (15–30 years old) were positive for '*molecular-FGS*' compared to older women (41–49 years old). These findings are consistent with earlier studies [23]. Older women are more likely to have chronic and long-standing *S. haematobium* egg deposition in the genital tract because of accumulated schistosome infection. Chronic granulomatous lesions persist even in the absence of current active infection [1,23]. However, women in *S. haematobium*-endemic areas are susceptible to

**(A)**

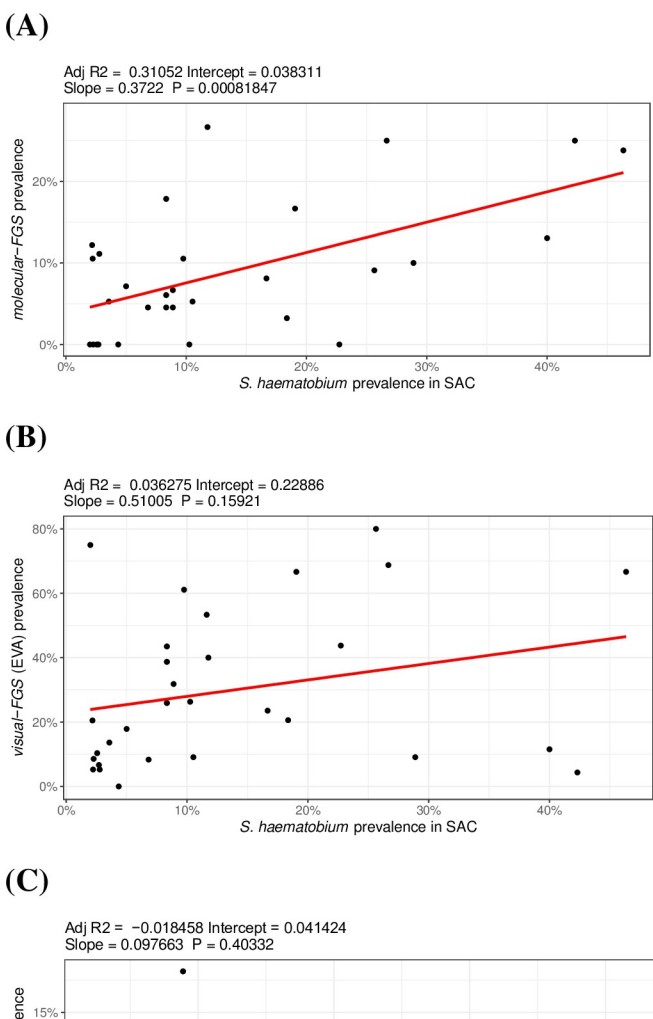

**(B)**

**(C)**

**Fig 3.** Village level prevalence of (A) '*Molecular-FGS*', (B) '*Visual-FGS*' by EVA MobileODT hand-held colposcopy, and (C) '*Visual-FGS* by Smart-Scope assessment (C) in women enrolled in the MORBID-FGS study vs village level prevalence of urine egg-positive *S. haematobium* in school aged children (SAC) enrolled in the main MORBID study.

schistosome infection from a young age and younger women often present a higher intensity of *S. haematobium* infection, and higher rates of schistosome DNA retrieval from the genital tract [1,23]. Early FGS lesions may be harder to diagnose visually in this age group [1,23].

Women with urinary schistosome infection diagnosed by microscopy were more likely to have '*molecular-FGS*' compared to women without. Microscopy and molecular diagnosis are highly specific for the diagnosis of schistosomiasis. Therefore, the correlation highlights the high specificity of '*molecular-FGS*' as a diagnostic method [3]. Findings contrast with previous work which found a variation in the association between *molecular-FGS* and urinary *S.*

*haematobium* infection detected by urine microscopy across five different study populations [7]. These contradictory findings and the variation in associations based on the study's inclusion criteria indicate a need for single and standardized procedure of sample collection and analysis.

No significant association was found between urinary *S. haematobium* status and *'visual-FGS'*. This poses a challenge to obtain accurate FGS burden assessments based on urinary egg-positivity alone. Review of colposcopic images is time consuming and requires expertise. Further, a recent study in Zambia revealed a low-level correlation between two expert reviewer readings [11]. *'Visual FGS'* diagnosis remains useful to detect FGS-related morbidity at an individual level, albeit not as specific as other diagnostics, however it is not scalable for population-based screening [11]. Hand-held colposcopy offers a closer to the user option and can be operated by midwives [2]. This study used one of the highest rated devices for FGS diagnosis and found fair level of agreement between different devices tested (EVA MobileODT vs Smart-Scope–Cohen's Kappa = 0·09, p-value = 0·02) [24].

*'Molecular-FGS'* was a strong predictor of *'visual-FGS'*, when adjusting for significant confounders. Schistosome DNA retrieval from the genital tract could be used as a proxy marker for FGS-related morbidity in some women. Molecular testing of genital samples for FGS screening has been previously piloted in research settings across SSA countries and showed high diagnostic accuracy [2,12,13]. However, cost, and high-technology laboratory needs limit its scalability. Genital self-sampling for community-based FGS surveillance at scale coupled with isothermal diagnostics (RPA) has been proposed as a potentially cheaper alternative to PCR methods [2,25]. As in other studies [2,23], our results showed a higher percentage of younger women with high *'molecular*-FGS' [25] suggesting that an age-specific target (younger groups [15–20]) for molecular screening could be useful for survey sampling.

Urinary, genital and, SRH self-reported signs and symptoms were not significantly associated with FGS (neither visual nor molecular). This differed from previous studies which report a significant association of FGS with a range of gynaecological and SRH manifestations [3,26]. Results from our analysis could be limited by the lack of data available to confirm STI status, which can confound the signs and symptoms of FGS [1,27]. This study attempted to adjust for previous STI history and found a negative association with *'visual-FGS'*. However, these results should be interpreted carefully as the history of STI was self-reported, making the results prone to recall bias. Moreover, self-reported infection history likely underestimates the true burden of disease since women were likely diagnosed and treated using a syndromic approach [28]. Future research should screen for STI in addition to FGS to further understand the STI context in *S. haematobium* endemic countries and explore the interplay between these and FGS [29].

Our study found that village level prevalence of urinary *S. haematobium* in SAC was correlated with *'molecular-FGS'* diagnosis but not with *'visual-FGS'*. Importantly, the prevalence of *'molecular-FGS'* was high (between 10.5% - 17.8%) in some low-prevalence villages for *S. haematobium* in SAC (below 10%, as per WHO guidelines) [21]. WHO recommends annual preventive chemotherapy with praziquantel in all age groups in endemic communities with schistosomiasis prevalence of 10% or higher [21]. Yet, treatment programmes do not target young girls and women in low prevalence villages (<10%) who are still at risk of FGS [21]. As such, control programs based on the SAC infection prevalence overlook the diagnosis and treatment of FGS in women living in endemic settings. Ideally, an affordable FGS molecular testing could be offered to young women and girls living in endemic settings regardless of the *S. haematobium* prevalence in SAC.

This study has some limitations. Highly sensitive and specific *Schistosoma* diagnostic assays such as the circulating anodic antigen (CAA) were not available to refine the infection-

morbidity and diagnostic accuracy correlations [30]. Active schistosomiasis relied on the diagnosis of urinary *S. haematobium* eggs, which has poor sensitivity for low intensity infections. Further, this study did not include histological examination of cervical biopsies limiting the reference comparison of '*visual-FGS*' and '*molecular-FGS*' as neither diagnostic methods provide perfect diagnostic accuracy.

Overall, the paucity of evidence on the burden of FGS in *S. haematobium* endemic countries highlights the need to conduct further burden of disease studies in areas with different endemicity levels. The high correlation between '*visual FGS*' and '*molecular FGS*' suggests that prevalence estimation can shift from using clinic-based colposcopy to implementation of scalable and field-deployable age-targeted molecular diagnostic methods, integrated in community-based screening algorithms [8]. In addition, with the promotion of the United Nations' SDGs, more attention will now be placed on women's health including sexual and reproductive health conditions [26]. Given the growing evidence on the associations between FGS, HIV, and cervical pre-cancer, screening and treatment for FGS may provide additional opportunity to reduce the burden of these potentially lethal but treatable conditions [4].

## Supporting information

**S1 STROBE Checklist. STROBE statement checklist of items that were included in the manuscript.**
(DOCX)

**S1 Text. Study design of the Morbidity Operational Research for Bilharzia Implementation Decisions (MORBID) study.**
(DOCX)

**S2 Text. Sample participants and sample size for the *parent* MORBID study.**
(DOCX)

**S3 Text. Randomized sampling procedures for the *parent* MORBID study.**
(DOCX)

**S4 Text. Sample selection procedure and randomization for the MORBID-FGS sub-study.**
(DOCX)

**S5 Text. Diagnosis of urinary *Schistosoma (S.) haematobium* by urine filtration.**
(DOCX)

**S6 Text. Genital DNA isolation and PCR.**
(DOCX)

**S1 Fig. Dacron swab used by the midwife at the study clinic for collection of cervicovaginal swabs.**
(TIF)

**S2 Fig. Self-reported urinary, genital, and sexual and reproductive health (SRH) signs and symptoms across the overall study population (n = 950), by '*visual-FGS*' status by EVA MobileODT (N = 880) and '*molecular-FGS*' by genital PCR (N = 899).**
(TIF)

**S1 Table. Baseline socio-demographic characteristics, water contact information, and history of urinary, genital, and sexual and reproductive health (SRH) signs and symptoms by FGS status diagnosed by colposcopic images ('*Visual-FGS*') and molecular methods**

(*'Molecular-FGS'*).
(DOCX)

**S2 Table. Baseline socio-demographic characteristics, water contact information, and history of urinary, genital, and sexual and reproductive health (SRH) signs and symptoms for 950 women across the study population and by the two study districts (Chikwawa and Nsanje) in Southern Malawi.**
(DOCX)

**S3 Table. Self-reported symptoms across age groups (n = 950).**
(DOCX)

**S4 Table. Self-reported symptoms across by district.**
(DOCX)

**S5 Table. Self-reported symptoms across the overall study population and by FGS status diagnosed by colposcopy (n = 880) and genital PCR (n = 899).**
(DOCX)

**S6 Table. Self-reported symptoms by FGS typical cervicovaginal lesions detected using the EVA MobileODT colposcope.**
(DOCX)

**S7 Table. Distribution of *'visual-FGS'* (cervical lesion detected by EVA MobileODT) positives (N = 241) by *'molecular-FGS'* status (parasite DNA detection by PCR).**
(DOCX)

**S8 Table. Distribution of *'visual-FGS'* by EVA MobileODT colposcopy and *'molecular-FGS'* by PCR of genital swabs across age groups.**
(DOCX)

**S9 Table. FGS typical cervical lesions observed using the Smart-scope colposcope.**
(DOCX)

**S10 Table. Generalized linear mixed models (GLMM) parameter estimates for association between *'molecular-FGS'* and *S. haematobium* infection status by urine filtration (n = 537).**
(DOCX)

**S11 Table. Generalized linear mixed models (GLMM) parameter estimates for association between *'visual-FGS'* status by EVA Mobile ODT and *S. haematobium* infection by urine filtration (n = 544).**
(DOCX)

## Acknowledgments

We would like to thank the study participants and the tireless study nurses Maria Zuze, Teresa Nchembe, Dorothy Ndovi, Alice Chisal who worked under challenging conditions during the COVID-19 epidemic. In memoriam of Dr. Bagrey Ngwira, an inspirational figure for this work. We would also like to thank Professor Charlie King for his input on the manuscript.

## Author Contributions

**Conceptualization:** Amaya L. Bustinduy.

**Data curation:** Olimpia Lamberti, Sekeleghe Kayuni, Bagrey Ngwira, Neerav Dhanani.

**Formal analysis:** Olimpia Lamberti, Neerav Dhanani.

**Funding acquisition:** Amaya L. Bustinduy.

**Investigation:** Dingase Kumwenda, Els Wessels, Lisette Van Lieshout.

**Project administration:** Sekeleghe Kayuni, Themba Mzilahowa.

**Resources:** Varsha Singh, Veena Moktali, Fiona M. Fleming, Amaya L. Bustinduy.

**Supervision:** Amaya L. Bustinduy.

**Visualization:** Olimpia Lamberti, Neerav Dhanani.

**Writing – original draft:** Olimpia Lamberti.

**Writing – review & editing:** Olimpia Lamberti, Sekeleghe Kayuni, Dingase Kumwenda, Varsha Singh, Veena Moktali, Neerav Dhanani, Els Wessels, Lisette Van Lieshout, Fiona M. Fleming, Themba Mzilahowa, Amaya L. Bustinduy.

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
