## [Decision Letter · Decision Letter 0]

25 Mar 2024

Dear Miss Lamberti,

We are pleased to inform you that your manuscript 'Female genital schistosomiasis burden and risk factors in two endemic areas in Malawi nested in the Morbidity Operational Research for Bilharziasis Implementation Decisions (MORBID) cross-sectional study.' has been provisionally accepted for publication in PLOS Neglected Tropical Diseases.

Best regards,

Uwem Friday Ekpo, PhD

Academic Editor

Uriel Koziol

Section Editor

Reviewer's Responses to Questions

**Key Review Criteria Required for Acceptance?**

**Methods**

-Are the objectives of the study clearly articulated with a clear testable hypothesis stated?

-Is the study design appropriate to address the stated objectives?

-Is the population clearly described and appropriate for the hypothesis being tested?

-Is the sample size sufficient to ensure adequate power to address the hypothesis being tested?

-Were correct statistical analysis used to support conclusions?

-Are there concerns about ethical or regulatory requirements being met?

Reviewer #1: The study methodology is well laid out and the design is appropriate. The study population has been adequately described and is appropriate for the study objectives to be measured against. The sample size is fine, it is rather a challenge at times to get an appropriate sample when investigating FGS, while this is a relevant disease, the exact extent is difficult to measure and FGS is a diagnostic challenge. The statistical methods are on point. The methods for ethical approval have been stated.

Reviewer #2: Are the objectives of the study clearly articulated with a clear testable hypothesis stated? Yes

Is the study design appropriate to address the stated objectives? Yes

Is the population clearly described and appropriate for the hypothesis ? Yes

Is the sample size sufficient to ensure adequate power to address the ? Yes

Were correct statistical analysis used to support conclusions? Yes

Are there concerns about ethical or regulatory requirements being met? No

**Results**

-Does the analysis presented match the analysis plan?

-Are the results clearly and completely presented?

-Are the figures (Tables, Images) of sufficient quality for clarity?

Reviewer #1: The analysis matches the plan. The results are well presented, only ensure that there is consistency in reporting the percentage and raw score for results- see line 360-

The tables and figures are sufficiently clear and of a good quality.

Reviewer #2: -Does the analysis presented match the analysis plan? Yes

-Are the results clearly and completely presented? Yes

-Are the figures (Tables, Images) of sufficient quality for clarity? Yes

**Conclusions**

-Are the conclusions supported by the data presented?

-Are the limitations of analysis clearly described?

-Do the authors discuss how these data can be helpful to advance our understanding of the topic under study?

-Is public health relevance addressed?

Reviewer #1: The conclusions are supported well by the data. Limitations are discussed. The public health relevance is adequately discussed.

Reviewer #2: -Are the conclusions supported by the data presented? Yes

-Are the limitations of analysis clearly described? Yes

-Do the authors discuss how these data can be helpful to advance our understanding of the topic under study? Yes

-Is public health relevance addressed? Yes

**Editorial and Data Presentation Modifications?**

Reviewer #1: Please consider adding an "s" to the word "setting" on line 86.

Reviewer #2: (No Response)

**Summary and General Comments**

Reviewer #1: This is an interesting study and provides valuable insight into a disease that many women from Schistoma endemic areas are at risk for, and a disease that is a diagnostic challenge. The study compared several methods to diagnose FGS, including urine microscopy and Schistosoma PCR which could be deemed proxy methods as well as direct visualization techniques for FGS. It is of interest that there was an increased prevalence of visual FGS among older women, this could be attributed to advanced stages of FGS in this population as they could have been infected in early childhood and have chronic FGS that is more easily manifested. Of concern though is the extent of FGS in younger and other women who may not yet clinically manifest. A limitation which has been noted by the authors, however, is worth emphasizing is the lack of complimentary diagnostic screening for other entities like HPV, SIL and other lesions that form the differential diagnosis for FGS. While the authors mention challenges with taking cervical biopsies, other less invasive alternatives such as HPV-DNA testing or even LBC could have been considered.

Reviewer #2: Fantastic layout with easy to assess tables.

PLOS authors have the option to publish the peer review history of their article (what does this mean?). If published, this will include your full peer review and any attached files.

Reviewer #1: No

Reviewer #2: No

---

## [Editor Report · Acceptance letter]

30 Apr 2024

Dear Miss Lamberti,

We are delighted to inform you that your manuscript, "Female genital schistosomiasis burden and risk factors in two endemic areas in Malawi nested in the Morbidity Operational Research for Bilharziasis Implementation Decisions (MORBID) cross-sectional study.," has been formally accepted for publication in PLOS Neglected Tropical Diseases.

Best regards,

Shaden Kamhawi

co-Editor-in-Chief

Paul Brindley

co-Editor-in-Chief
